# Noncoding RNAs Controlling Oxidative Stress in Cancer

**DOI:** 10.3390/cancers15041155

**Published:** 2023-02-10

**Authors:** Paul Holvoet

**Affiliations:** Division of Experimental Cardiology, KU Leuven, 3000 Leuven, Belgium; paul.holvoet@kuleuven.be

**Keywords:** oxidative stress, immune response, cancer, noncoding RNAs, microvesicles

## Abstract

**Simple Summary:**

Mitochondria in cancer cells produce reactive oxygen species, inducing a vicious cycle between oxidative stress, genomic instability, and cancer development. However, too high oxidative stress kills tumor cells. Cancer cells protect themselves directly by increasing antioxidant expression. In addition, shifting the phenotype of immune cells from ant-oncogenic to pro-oncogenic reduces reactive oxygen levels within the tumor microenvironment to maintain tumor growth. This review shows how noncoding RNAs may regulate mitochondrial function, antioxidant expression, and immune cell reprogramming. Interestingly, noncoding RNAs in microvesicles secreted by mesenchymal stem cells, cancer-associated fibroblasts, and cancer cells contribute to this immune cell reprogramming. Further research on the role of noncoding RNAs in the communication between cell types in the tumor microenvironment is warranted.

**Abstract:**

Mitochondria in cancer cells tend to overproduce reactive oxygen species (ROS), inducing a vicious cycle between mitochondria, ROS, genomic instability, and cancer development. The first part of this review deals with the role of noncoding RNAs in regulating mitochondrial ROS production and the expression of antioxidants in cancer cells, preventing the increase of ROS in the tumor microenvironment. In addition, cytotoxic T and natural killer cells release high levels of ROS, inducing cell death, while anti-immune regulatory T cells, tumor-associated M2 macrophages, and myeloid-derived suppressor cells, at least at the initial stage of tumor growth, release low levels of ROS supporting tumor growth. Therefore, this review’s second part deals with noncoding RNAs’ role in regulating the metabolic reprogramming of immune cells about ROS release. Furthermore, the enrichment of noncoding RNAs in microvesicles allows communication between cell types in a tumor and between a tumor and tumor-adjacent tissues. Therefore, the third part illustrates how noncoding RNA-containing microvesicles secreted by mesenchymal stem cells and primary tumor cells may primarily aid the shift of immune cells to a pro-oncogenic phenotype. Conversely, microvesicles released by tumor-adjacent tissues may have the opposite effect. Our review reveals that a specific noncoding RNA may affect oxidative stress by several mechanisms, which may have opposite effects on tumor growth. Furthermore, they may be involved in mechanisms other than regulating oxidative stress, which may level out their effects on oxidative stress and tumor growth. In addition, several noncoding RNAs might share a specific function, making it very unlikely that intervening with only one of these noncoding RNAs will block this particular mechanism. Overall, further validation of the interaction between noncoding RNAs about cancer types and stages of tumor development is warranted.

## 1. Introduction

In 2000, Hanagan and Weisberg defined six hallmarks of cancer: self-sufficiency in growth signals, insensitivity to growth-inhibitory signals, evasion of programmed cell death, unlimited replicative potential, sustained angiogenesis, and tissue invasion and metastasis [1]. The controlled release of reactive oxygen species (ROS) is crucial for tumor growth in hypoxia [2]. Mitochondria in tumor cells tend to overproduce ROS, particularly by NADPH oxidases (NOX). ROS induce genomic instability, and mitochondrial and nuclear DNA mutations caused by oxidative damage impair the oxidative phosphorylation process, resulting in further mitochondrial ROS production, completing the vicious cycle between mitochondria, ROS, genomic instability, and tumor growth [3]. ROS promotes cellular proliferation by mitogen-activated protein kinases (MAPK), such as extracellular-regulated kinase 1/2 (ERK1/2), nuclear factor erythroid 2-related factor 2 (NRF2), kelch-like protein 19 (Keap1), Ras, Raf, c-Jun N-terminal kinase (JNK) and MYC; evasion of apoptosis by a phosphatidylinositol-3 kinase (PI3K)/AKT; epithelial–mesenchymal transition (EMT) by SMAD, activated by tumor growth factor β (TGF-β), SNAIL, E-cadherin, and β-catenin; tissue invasion and metastasis by metalloproteinases (MMPs); and angiogenesis by vascular endothelial growth factor (VEGF) and angiopoietin [4].

Tumor cells release ROS into the tumor microenvironment and are detected by adjacent fibroblasts (CAFs), initiating the onset of stromal oxidative stress, autophagy, and mitophagy due to activating HIF-1α and VEGF that will ultimately contribute to angiogenesis. Moreover, CAFs also release MMPs and IL-6, IL-10, TGF-β, C-C motif chemokine ligand 22 (CCL)-2, and CCL5, which stimulate tumor growth and block the natural immune response against cancer, as described above.

In addition, ROS secreted by pro-tumorigenic CAFs will affect tumor growth by regulating NO production. NO has broad and sometimes dichotomous roles in cancer; the effects of NO in tumors depend on the type and localization of NOS isoforms, concentration and duration of NO exposure, and cellular sensitivity to NO. For example, pro-tumorigenic CAFs upregulate nitric oxide synthases (NOS)-1 with nuclear factor erythroid 2-related factor 2 (NRF2 and HIF1A in the tumor microenvironment [5]. Then, NOS1 translocation in the mitochondria induced sirtuin (SIRT)-3 that eliminates ROS from the mitochondria and increases apoptosis resistance, for example, in colon cancer [6]. However, NO generated by the inducible NOS (iNOS or NOS2) and the co-expression of iNOS with mitochondrially encoded cytochrome c oxidase II (MT-COII or COX2) may lead to a particularly aggressive cancer phenotype of breast, liver, colon, pancreatic ductal adenocarcinoma, and hepatocellular cancers [7].

Ultimately, ROS levels in tumor-related cells depend on the tumor’s antioxidant capacity, which counterbalances ROS increase. This capacity largely depends on NRF2 [8]. Indeed, the activation of NRF2 requires that ROS activates tyrosine kinases, which dissociate the NRF2 from its inhibitor, Keap1 [9]. This dissociation allows the nuclear import of NRF2 to activate antioxidant peroxiredoxin-1 (PRDX1), peroxiredoxin-like 2A (PRXL2A), glutathione peroxidase (GPX), thioredoxin-1 (TXN1), catalase (CAT), and superoxide dismutase (SOD) [10]. In addition, NRF2 interacts with peroxiredoxin 5 (PRDX5), promoting the expression of NAD(P)H: quinone oxidoreductase 1 (NQO1) [11].

Furthermore, in 2011, Hanagan and Weisberg added two more hallmarks: evasion of immune destruction and tumor-promoting inflammation [12]. The composition of the immune microenvironment is diverse, including various populations of T cells, dendritic cells, natural killer cells, myeloid-derived suppressor cells (MDCSs), and macrophages [13]. Adapting the immune response associated with metabolic reprogramming is crucial in regulating oxidative stress. In particular, diminishing the activity of cytotoxic T cells, augmenting the activity of regulatory T cells (T_reg_) and myeloid-derived suppressor cells (MDSCs), and preventing the M2 into M1 macrophage differentiation limits ROS release. Hypoxia and hypoxia-inducible factor 1, alpha (HIF1A) through Wnt/β-catenin and ROS ablates cytotoxic CD8^+^ T-cell function. Furthermore, activation of the nuclear factor kappa B (NF-κB) subunit 1 induces apoptosis in T cells [14]. In addition, NADPH oxidase 2 (NOX2)-derived ROS from MDSCs facilitate cancer metastasis by downmodulating natural killer (NK) T cell function. Conversely, inhibition of NOX2 restores interferon (IFN)-γ-dependent cell death [15].

ROS secreted by MDSCs and interleukin (IL)-4, IL-10, and IL-13 secreted by T helper (Th)-2 cells stimulate M2 tumor-associated macrophage (TAM) proliferation [16]. In addition, MDSCs and TAMs secrete IL-10, macrophage migration inhibitory factor (MIF), and transforming growth factor-β (TGF-β) to hamper NK cell function [17]. Further, TAMs suppress the anti-tumor responses of dendritic cells (DCs). In addition, MDSCs and M2 TAMs induce T_reg_ cells by activating the TGF-β/SMAD signaling pathway [18]. In contrast, iNOS secreted by CAFs [19] and IFN-γ secreted by Th1 cells cause M2 to M1 macrophage polarization associated with mitochondrial dysfunction, increased ROS release, and apoptosis [20]. In addition, granulocytic MDSCs in growing tumors exposed for a longer time to hypoxia may produce high levels of ROS, inducing senescence in T_reg_ cells and apoptosis [21]. Finally, evidence suggests an essential cross-talk between CAFs and immuno-inhibitory cell types. For example, CAFs in breast cancer may attract CD4^+^ CD25^+^ T cells by secretion of C-X-C motif chemokine ligand 12 (CXCL12) and promote CD4^+^ T-cell differentiation into FOXP3-positive T_reg_ cells or contribute to immunosuppressive M2-like TAM induction in breast cancer and oral squamous cell carcinoma [22]. T cell dysfunction may occur when effector T cells are exposed to ROS produced by MDSCs and Tregs, which are more resistant to oxidative stress due to their increased antioxidant systems. High levels of extracellular ROS can disrupt antigen presentation between T cells and DCs and can affect tumor antigen recognition by T cells [23].

Because immune cells determine ROS levels in the tumor environment to a large extent, they are the main focus of this review. In particular, we focused on the role of noncoding RNAs in regulating oxidative stress. Upon exposure to stress conditions, their expression changes more rapidly than that of protein transcription factors, and a change in their expression is reversible in contrast to that of inherited genome mutations [24]. Herein, we make an effort to include information about possible targets. However, one has to be aware that studies often stress one target’s importance without considering other targets, disregard the nonspecific effects of silencing experiments, and do not include reconstitution experiments. Moreover, a single noncoding RNA is not specific for one target but may affect several targets within one or several crucial pathways in one or several cell types within one or several tissues. Therefore, one must combine data from many studies to capture their association with a disease. In addition, several noncoding RNAs may affect the same target, influencing each other’s expression or action. Therefore, overall we prefer to illustrate in the enclosed figures changes in the cell’s phenotype associated with a change in the expression of a noncoding RNA rather than rigorously confining a particular target.

Furthermore, phenotypic changes in immune cells related to oxidative stress are not necessarily due to noncoding RNAs secreted by immune cells but may be transferred to them by the other tumor cells via microvesicles. Indeed, microvesicles facilitate the communication between different tissues and cell types within these tissues [25]. Moreover, noncoding RNAs may be secreted in extracellular vesicles or exosomes, contributing to the communication between several cell types within one tissue or between several tissues. Tumor cells and TAMs interact through extracellular vesicles in various cancers, such as pancreatic ductal adenocarcinoma, gastric cancer, breast cancer, ovarian cancer, colon cancer, glioblastoma, hepatocellular cancer, and lung cancer. They may be enriched explicitly in noncoding RNAs [26].

The graphical abstract illustrates how noncoding RNAs could regulate oxidative stress in tumors by affecting mitochondrial function, expression of antioxidants, reprogramming of immune cells, and exchange of microvesicles.

## 2. Noncoding RNAs and Mitochondrial Dysfunction

MiR-9-5p directly downregulated pyruvate dehydrogenase kinase 4, located in the matrix of mitochondria, promoting cell proliferation, invasion, and migration, enhancing mitochondrial activity and energy metabolism, reducing ROS, and suppressing apoptosis in hepatocellular cancer cells [27]. By attenuating the decline in cellular mitochondrial membrane potential, miR-21 suppressed cytochrome c release to the cytoplasm, followed by a decrease in the activity of intracellular caspase-9 and caspase-3, reflecting impaired mitochondrial-mediated proapoptotic pathway, ultimately reducing intracellular ROS [28]. The decrease in ROS will lead to a reduction in NF-κB that would otherwise activate many inflammatory factors, such as TNF-α, COX2 (or MT-COII), and iNOS [29]. Depletion of HOX transcript antisense RNA (HOTAIR) induced mitochondrial-related cell death pathways related to BCL2, BAX, caspase-3, cleaved caspase-3, and cytochrome c [30]. The nuclear-encoded metastasis-associated lung adenocarcinoma transcript 1 (MALAT1), enriched in the mitochondria of hepatoma cells, interacted with multiple loci on mitochondrial DNA, including MT-COII (or COX2). MALAT1 knockdown altered mitochondrial structure, impaired oxidative phosphorylation (OXPHOS), decreased ATP production, reduced mitophagy, decreased mtDNA copy number, and activated mitochondrial apoptosis [31]. The PVT1 oncogene (PVT1) was among the genes amplified in breast cancer cells related to mitochondria-regulated nuclear gene expression helps breast cancer cells survive and proliferate [32]. The urothelial cancer-associated 1 (UCA1) enhanced mitochondrial function and bladder cancer cell viability through the UCA1/miR-195/ARL2 axis [33] (Figure 1).

An increase of tumor-suppressive let-7 family and miR-34a-5p led to a decrease of MYC and perturbation of mitochondrial function. Knockdown of the let-7 family or miR-34a-5p restored MYC levels and mitochondrial function [34]. MiR-30a promoted mitochondria-dependent intrinsic apoptosis [35]. MiR-124 induced apoptosis via the intrinsic mitochondrial pathway in human oral squamous cell carcinoma cells [36]. MiR-125 targeted cancer cells and intracellular mitochondria, altering cellular bioenergetics, lipid, and glucose metabolism, inducing apoptosis in human tongue squamous carcinoma cells [37]. MiR-128 suppressed SIRT1 expression, promoting the production of ROS and apoptosis in TRAIL-treated colorectal cancer cells [38]. The mitochondrially localized growth arrest-specific transcript 5 (GAS5) could disrupt mitochondrial membrane potential and promote BAX, BAK, cleaved-caspase 3, and cleaved-caspase 9 expressions in epithelial ovarian can cells [39] and inhibit associated mitochondrial metabolic enzymes in breast cancer cells in response to nutrient stress [40] (Figure 1).

## 3. Noncoding RNAs and Antioxidant Activity

MALAT1 in myeloma [41] and PVT1 in breast cancer promoted NRF2 protein stability by inhibiting Keap1 binding to NRF2 [42]. UCA1 targets miR-495, inducing NRF2 in non-small-cell lung cancer [43]. Taurine upregulated 1 (TUG1) post-translationally potentiates the NRF2 effect in urothelial cancer [44]. LncRNA SNHG14 is crucial to retaining NRF2 activity in breast cancer [45]. HOXA11-AS in oral squamous cell carcinoma [46] and NEAT1 in breast cancer [47] positively regulated NQO1. NEAT1 induces GPX in melanoma [48], while XIST induces NRF2, GPX, and SOD2 in non-small-cell lung cancer [49]. MiR-17 reduced superoxide dismutase 2 (SOD2 or MnSOD) and GPX in prostate cancer cells [50]. Finally, miR-155 caused inhibition of FOXO3A, leading to a decrease of SOD2 and CAT and enhanced ROS generation in pancreatic cancer [51] (Figure 2).

## 4. Immune Cells and ROS Generation

### 4.1. T Cells

Cancer cells evade immune suppression by downregulating the activity of cytotoxic CD8^+^ T cells and NK cells and increasing Treg cells’ activity [52]. Hypoxia activates Wnt/β-catenin and reduces cytotoxic CD8^+^ T-cells, decreasing innate immunity involving DCs and NK cells [53]. In addition, hypoxia induces HIF1A, ADAM metallopeptidase domain 10 (ADAM10), and MHC class I polypeptide-related sequence A (MICA), decreasing NK cell-mediated cell death depending on IFN-γ and tumor necrosis factor (TNF-α) [54]. Moreover, IL-4 synergistically enhances the IL-2- and IL-12-induced IFN-γ expression [55], further stimulated by platelet-derived growth factor (PDGF)-DD [56]. In addition, TGF-β inactivates NK cells [57] (Figure 3).

DC-derived IL-12 differentiates Th1 cells [58]. Th1-derived IL-2 maintains T_reg_ cells [59], and granulocyte–macrophage colony-stimulating factor (GM-CSF) [60] induces the recruitment and differentiation of MDSCs. In contrast, IFN-γ promotes M1 macrophage polarization and cytotoxic inflammation [16]. Th2 cells produce IL-4, IL-10, and IL-13, stimulating M2 TAM differentiation [16,61]. Th17 cells cause immunosuppression and promote tumor growth by IL-17, inducing angiogenesis and recruiting MDSCs. In contrast, Th17 cells cause anti-tumoral immunity by recruiting DCs, CD4^+^ T, and CD8^+^ T cells, activating CD8^+^ T and Th1 cells [62], producing IFN-*γ* and TNF-α [63].

T_reg_ cells are FOXP3-positive CD4^+^ T cells strongly inhibiting anti-tumor immune responses [64]. They inhibit CD80 and CD86 expressed by DCs and directly kill effector T cells [65]. IL-10 and TGF-β secreted by MDSCs induce the differentiation of T_reg_ cells [66]. IL-10 and IL-23 secreted by macrophages reinforce this differentiation, promoting IL-10 and TGF-β expression, thereby suppressing tumor cell killing [67].

In aggregate, the activation of cytotoxic T cells, NK cells, and dendritic cells increase ROS. Conversely, the activation of T_reg_ and Th2 cells decreases ROS. The activation of Th1 cells increases ROS by activating dendritic cells but decreases ROS by activating T_reg_ cells. The Th17 cells could also have divergent effects by activating MDCs and Th1 cells (Figure 3).

### 4.2. Noncoding RNAs and T Cells

MYC expression is deregulated in various cancer types. In breast cancer, MYC is overexpressed in 30–50% of high-grade tumors [68]. Inactivation of MYC oncogene sustained regression of invasive liver cancers [69]. MYC is frequently overexpressed in both sporadic and colitis-associated colon adenocarcinomas [70]. Furthermore, MYC is an adverse prognostic marker in colorectal cancer [71]. Multiple myeloma is considered a plasma cell malignancy associated with MYC deregulation [72]. MYC exerts a context- and cell-dependent function. For example, MYC acts early in T cell activation [73]. However, amplification of *MYC* alone is insufficient for tumor development in vivo; co-amplification of the *PVT1* gene downstream of the *MYC* gene is required. In colorectal cancer samples, *PVT1* gene expression increased 10-fold, associated with an enrichment of CD8^+^ T cell subsets in colorectal cancer lesions. In addition, the expression of *PVT1* transcripts with the open reading frame in target T cells correlated with IFN-γ production [74]. MiR-150 hampered the activation of CD8^+^ T cells [75], and overexpression of miR-150 reduced inducible NKT cells [76]. However, the upregulation of PVT1 in hepatocellular carcinoma silenced miR-150, leading to the overexpression of the hypoxia-inducible protein 2 [77], which induced a cytotoxic T-cell response in patients with metastatic renal cell carcinoma [78]. MiR-155 is needed for CD8^+^ T cell responses to cancer cells. In the absence of miR-155, the number of effector CD8^+^ T cells was reduced, and miR-155-deficient CD8^+^ T cells were ineffective at controlling tumor growth by the upregulation of suppressor of cytokine signaling-1, causing defective cytokine signaling through signal transducer and activator of transcription (STAT)-5 in melanoma [79]. Furthermore, NEAT1 or LDHA knockdown promoted the secretion of CD8^+^ T-lymphocyte factors, including TNF-α and IFN-γ, enhancing the anti-tumor effects [80]. In contrast, UCA1 attenuated the killing effect of cytotoxic CD8^+^ T cells by targeting miR-148a [81] (Figure 3).

Hypoxia-induced GAS5 promotes the anti-tumor effect of NK cells by sponging miR-18a in gastric carcinoma, increasing the secretion of IFN-γ, TNF-α [82], and targeting miR-544, upregulating the RUNX family transcription factor 3 in liver carcinoma [83]. In contrast, increased levels of miR-20a in ovarian cancer tumor cells may indirectly suppress NK cell cytotoxicity by downregulating MICA/B expression [84].

Furthermore, miR-155 is induced upon T-cell activation and promotes Th1 differentiation when over-expressed in activated CD4^+^ T cells. Antagonism of miR-155 leads to the induction of IFN-gamma receptor alpha-chain [85]. MiR-17 and miR-19b control Th1 responses, for example, by inducing the release of IFN-γ and suppressing T_reg_ cell differentiation by hampering FOXP3-mediated activation [86].

MiR-21 regulates the glycolysis of CD4^+^ T cells through the PTEN/PI3K/AKT pathway to accelerate the cell cycle, thereby facilitating CD4^+^ T cell polarization toward Th2 cells, releasing IL-13 [87]. Multiple myeloma is tightly dependent on the inflammatory bone marrow microenvironment, and IL-17-producing Th17 cells sustain multiple myeloma cell growth. However, early inhibition of miR-21 in naive T cells impaired Th17 differentiation [88]. MiR-130b overexpressed in diffuse large B-cell lymphoma was associated with Th17 cell activation with IL-17 release [89]. MiR-155, increased in cervical cancer tissues, inhibited the expression of target gene SOCS1, promoting the differentiation of Th17 cells and increasing IL-17 [90].

Hypoxia and MYC downregulate miR-34a and increase CCL22, recruiting T_reg_ cells in hepatocellular carcinoma [91]. Upon T cell receptor stimulation, CD4^+^ T helper lymphocytes release miR-containing extracellular vesicles. A significant increase in miR-21 and a significant reduction in miR-155 resulted in increased T_reg_ differentiation in pediatric acute lymphoblastic leukemia [92]. MiR-124 induced the STAT3 pathway, reversed the glioma cancer stem cell-mediated immunosuppression of T-cell proliferation, and induced FOXP3-positive Treg cells. Treatment of T cells from immunosuppressed glioblastoma patients with miR-124 induced marked effector response, including upregulation of IFN-γ and TNF-α [93]. The HOXA cluster antisense RNA 2 (HOXA-AS2) promotes the differentiation of T_reg_ cells by sponging miR-302a and upregulating lysine demethylase 2A and jagged 1 in glioma cells [94] (Figure 3).

## 5. MDSCs and Macrophages

MDSCs arise from monocytes or polymorphonuclear cells [95]. CXC motif chemokine ligand 2 (CXCL2) released by tumor cells, which in turn activates CXC chemokine receptor 2 (CXCR2), and IL-6, IL-8, and IL-1β recruit tumor-infiltrating MDSCs inducing epithelial–mesenchymal transition (EMT) and metastasis [96]. Cytotoxic CD8^+^ cells [97] and NK cells secrete IFN-γ and TNF-α that induce tumor cell growth arrest [56]. In addition, Th1-derived IFN-γ induces M1 macrophage polarization and cytotoxic inflammation [16]. IFN-y and IL-17 signaling are crucial canonical pathways involved in MDSC immunomodulation: IFN-y triggers the immunosuppressive properties of MDSCs, whereas IL-17 induces MDSC tumor invasion [98]. Th1-derived granulocyte–macrophage colony-stimulating factor (GM-CSF) [60] and IL-17 secreted by Th17 cells recruit and differentiate MDSCs [62]. However, IFN-γ-CD8^+^T cells [99] and M1 TAMS [100] secrete IL-12, reprograming MDSCs to enhance T cell-mediated tumor regression. IL-12 induced tumor-infiltrated CD45^+^ and proliferative IFN-γ-positive CD8^+^ T cells and reduced CD4^+^ FOXP3^+^ T_reg_ cells [101] (Figure 4).

TAMs derive from circulating monocytes recruited by chemokines (CCL-2, CCL-3, CCL-4, CCL-5, CCL-7, CCL-8, CXCL-12) and cytokines (VEGF, PDGF, M-CSF, and IL-10) [102]. Moreover, hypoxia polarizes macrophages toward an M2 phenotype through TGF-β [103]. MDSCs secrete IL-10 [104] and IL-6, which promote the polarization of monocytes/macrophages toward an M2 phenotype [105]. In addition, ROS secreted by MDSCs inhibit IFN-γ secretion and proliferation of cytotoxic CD8^+^ T cells and the differentiation of M1 macrophages and DCs11c^+^ CD11b^+^ DCs [106]. Th2 cells produce IL-4, IL-10, and IL-13, stimulating M2 TAM differentiation [16,61]. In contrast, IL-17 promotes the M2 to M1 TAM polarization, increasing ROS and inducing cell death [107].

TAMs produce growth-inducing hepatocyte growth factor, epidermal growth factor, TGF-β, and inflammatory IL-1β, IL-6, and TNF-α that can induce EMT in cancer cells. On the other hand, EMT facilitates M2 TAM polarization through GM-CSF, which further induces EMT in cancer cells [108]. In addition, TAMs highly express TGF-β and TNF-α, promoting EMT and cancer stemness [109]. A PD-L1 inhibitor blocked IL-13-induced M2 TAM polarization, EMT, and stemness [110]. Finally, M2 TAMs promoted tumor angiogenesis [111].

In contrast, LPS and IFN-γ switch M2 to M1 TAMs [20,112]. Finally, NO produced by CAFs appears to have a critical role in the reprogramming process of M2 to M1 phenotype and the tumor environment [113] and causes oxidative/nitrosative stress [114] (Figure 4).

## 6. Noncoding RNAs and MDSCs, and Macrophages

MiR-30a in B-cell lymphoma targeted suppressor of cytokine signaling 3 (SOCS3), activating JAK-STAT3 signaling and promoting MDSC differentiation and immunosuppression [115]. HOTAIR, highly expressed and associated with poor prognosis in hepatocellular carcinoma, promoted the secretion of CCL2 in hepatocellular carcinoma cells and increased the proportion of macrophages and MDSCs [116] (Figure 4).

Low miR-181b is associated with increased monocyte infiltration by inducing EGFR-dependent VCAM-1 expression mediated by the p38/STAT3 signaling in glioblastoma [117].

The inhibition of let-7 by Wnt-induced H19 facilitates M2 TAM differentiation in pulmonary carcinoma [118]. The anti-inflammatory activity of IL-10 depends at least partly on the induction of miR-146b in a STAT3-dependent manner; miR-146b modulates the TLR4 signaling pathway by dampening MyD88, IRAK1, and TRAF6 [119]. miR-223 suppresses the activation of the canonical and non-canonical NF-κB pathways, STATs, and NLRP3 inflammasome and, therefore, is a potent negative regulator of inflammation by suppressing neutrophil activation and macrophage M1 polarization [120]. The HOMER3 antisense RNA 1 (HOMER3-AS1) induced by Wnt/β-catenin upregulates HOMER3 and promotes M2 TAM polarization in hepatocellular carcinoma. Functional rescue experiments revealed that HOMER3-AS1 increased colony-stimulating factor-1 (CSF-1) expression and secretion [121].

In contrast, let-7d in the conditioned medium of renal cell carcinoma cells inhibited macrophage M2 polarization, reducing IL-10 and IL-13 [122]. Let-7b increased IL-12 in prostatic M1 TAMs [123]. MiR-16 induced IL-12 in M1 macrophages [124], whereas miR-128 increased IL-12 and decreased IL-6 and IL-10 [125]. IL-12 induces miR-155, which is crucial for the cytotoxicity of NK cells [126] and reprograms TAMs to pro-inflammatory M1 macrophages [127]. However, MIR155HG is expressed higher in M2 macrophages and lower in M1 and sponges miR-155 in melanoma [128]. In addition, the HOXA distal transcript antisense RNA (HOTTIP) induced polarization of M2 into M1 TAMs in neck squamous cell carcinoma. HOTTIP activated the TLR5/NF-κB signaling pathway by competitively sponging miR-19a-3p and miR-19b-3p. Furthermore, cancer cells expressing HOTTIP induced the polarization of both local M1 and M2 macrophages; however, M1 exosomes could reprogram local TAMs into M1 macrophages. More importantly, cancer cells expressing HOTTIP and M1 exosomes reeducated circulating monocytes to express the M1 phenotype [129] (Figure 4).

## 7. Role of Noncoding RNA-Containing Microvesicles

Microvesicles facilitate the communication between different tissues and cell types within these tissues [25]. For example, miR-9 secreted in exosomes from triple-negative breast cancer cells induces the differentiation of fibroblasts to cancer-associated fibroblasts, increasing cell motility [130]. Melanoma cell-secreted exosomes enriched in miR-155 induced the reprogramming of fibroblasts into CAFs, triggering a proangiogenic switch by promoting the expression of proangiogenic factors, including VEGFa, fibroblast growth factor 2, and MMP9, by directly targeting SOCS1 and activating JAK2/STAT3 [131]. Interestingly, 8-OHD provoked ROS overproduction by downregulating JAK/STAT, PI3K/AKT, and oxidative phosphorylation [132]. In addition, human melanoma-derived exosomes reprogramed human adult dermal fibroblasts, increasing aerobic glycolysis and decreasing oxidative phosphorylation via miR-155 and miR-210 [133] (Figure 5).

In addition, cancer cell exosome-derived miR-9 and miR-181a promote the expansion of MDSCs from granulocytes in breast cancer by activating the JAK/STAT signaling pathway via targeting SOCS3 and the protein inhibitor of activated STAT3, PIAS3 [134]. Tumor cells may inhibit M1 polarization and promote cancer growth by selective shuttling of miR-21 enriched microvesicles [135,136]. The cancer cell exosomal ELFN1 antisense RNA 1 (ELFN1-AS1) was high in patients with advanced osteosarcoma. In addition, overexpression of ELFN1-AS1 significantly promoted the proliferation, migration, and invasion of osteosarcoma cells, while knockdown of ELFN1-AS1 exhibited the opposite effects. Meanwhile, exosomal ELFN1-AS1 could be transferred from osteosarcoma cells to macrophages, promoting macrophage M2 polarization by sponging miR-138-5p and miR-1291, which may facilitate osteosarcoma progression [137]. In contrast, exosomes isolated from 4T1 breast cancer cells delivering a miR-33 mimic into IL-4 induced M2 macrophages converted them to the M1 phenotype as indicated by an increase in expression of M1 markers, including IRF5, NOS2, and CD86, and a decrease in M2 markers including ARG, Ym1, and CD206. Furthermore, the secretion of TNF-α and IL-1β decreased, while the secretion of IL-10 and TGF-β increased [138].

MiR-150 and miR-7 were required for mesenchymal stem cells to decrease NK cell activities and recruit T_reg_ cells [139]. In addition, miR-150 in lung cancer-secreted EVs reduced the expression of CD226 on NK cells, creating a favorable immunosuppressive microenvironment by releasing IL-10 [140]. Hypoxia-primed CAFs secrete exosomes enriched in miR-21-5p, promoting macrophage M2 polarization and reducing apoptosis. Injection in immunocompromized mice significantly increased tumor growth, cancer cell proliferation, intra-tumoral angiogenesis, and M2 polarization of macrophages [135].

IL-4-activated macrophages co-cultivated with breast cancer cells without direct secreted miR-223-enriched microvesicles promoting the invasion of breast cancer cells via upregulating Mef2c [141]. Its expression depends on the overexpression of the antioxidant enzyme SOD1 [142]. In addition, exosomes derived from hypoxic macrophages enhanced the malignant phenotype of epithelial ovarian cancer cells enriched in miR-223 that induced PI3K/AKT signaling, which is crucial for controlling oxidative stress [143].

However, M1 macrophage in inflamed tumor-adjacent tissues secrete microvesicles enriched in the let-7 and miR-17/92 families, together with miR-21, miR-126, miR-146, miR-155, and miR-223, are enriched in M1 macrophage-derived inflammatory microvesicles [25,144]. In addition, microvesicles secreted by immature dendritic cells (DCs) contain more miR-21, miR-221, and miR-222. Microvesicles secreted by mature DCs contain more miR-146 and miR-155 [145]. In particular, sorting miR-17, miR-21, miR-150, miR-155, miR-221-222, and miR-223 in microvesicles secreted by M1 macrophages in tumor-adjacent tissue may affect the phenotype of immune cells in the tumor’s microenvironment (Figure 5).

In addition, CAFs and cancer cells may secrete exosomes promoting the polarization of macrophages to an M1 (anti-tumoral profile), expressing more NOS2 or M2 (pro-tumoral profile) phenotype, expressing more arginase [146].

## 8. Discussion

We reviewed the role of noncoding RNAs in regulating oxidative stress in the tumor environment by controlling mitochondrial function, antioxidant defense, and reprogramming of immune cells. Overall, we considered three groups of noncoding RNAs (Table 1). The overexpression of miR-9, miR-20a, miR-21, miR-130b, miR-146b, miR-150, miR-223, HOTAIR, MALAT1, UCA1, and XIST supports oncogenic mechanisms. In contrast, the overexpression of let-7b, let-7d, miR-16, miR-34a, miR-125, miR-128, miR-155, GAS5, and HOTTIP supports anti-ocogenic mechanisms. However, miR-17-19, miR-30a, miR-124, NEAT1 and PVT1 exert pro- and anti-oncogenic functions.

Overall, we consider miR-9, miR-21, miR-155, and miR-181a as crucial noncoding RNAs because they are enriched in microvesicles that aid the communication between immune and cancer cells. Moreover, they play other essential roles in cancer biology. For example, induction of miR-9 maintains the mesenchymal stem-cell potential in cancer cells [147], induces EMT [148], and inhibits apoptosis [149]. MiR-9 promotes angiogenesis by VEGF [150]. In addition, miR-21 maintains stem cell markers OCT4 and CD133 [151], induces EMT [152], and inhibits apoptosis [153]. In contrast, TGF-β1-induced miR-155 may potentiate TGF-β-induced EMT [154]. MiR-181a/b induces apoptosis by targeting BCL2 [155].

In addition, the microvesicle-mediated transfer of miR-33, miR-150, and lncRNA ELFN1-AS1 may affect the reprogramming of immune cells in tumors. In addition, miR-33 inhibited EMT [156] and led to a cell cycle arrest by directly targeting cyclin-dependent kinase 6 (CDK6) and cyclin D1 (CCND1) [157] and elevated apoptosis-related protein expression in MCF-7 [158]. PVT1 silences miR-150, resulting in the overexpression of hypoxia-inducible protein 2, which promotes the immune escape from NK cells through an IL13/STAT3 signaling pathway [77]. Low expression of miR-150 is also associated with tumor invasion induced by CCL20 and IL22, which lowers anti-tumor immunity and protects stemness [159]. Hypoxia-induced ELFN1-AS1 potentiates stemness [160], promotes proliferation and invasion, and inhibits apoptosis [161].

However, we have to be aware that miR-17, miR-21, miR-150, miR-155, miR-221–222, and miR-223 enriched in microvesicles secreted by inflamed tumor-adjacent tissues also affect the behavior of tumor-associated immune cells. These miRs may be an essential link between metabolic diseases like obesity, diabetes, and non-alcoholic fatty liver disease, with cancer risk [26].

However, not all noncoding RNAs possibly interfering with oxidative stress in cancer have been mentioned. Ultimately, we made a selection of well-documented noncoding RNAs. This selection explains why circular RNAs have not been discussed. Furthermore, the quality of this review paper depends on that of the original studies, and shortcomings in the original studies will affect the content of this review. First, functional and clinical studies are mostly limited to a single noncoding RNA, although noncoding RNAs function in networks, as illustrated in this review. Furthermore, the outcome of silencing a single lncRNA is attributed chiefly to one miR’s effect, although no broader omics search is performed to unravel all potentially involved miRs. In addition, the nonspecific effects of silencing experiments are mostly disregarded, and often reconstitution experiments are not performed. These shortcomings may explain contrasting results. In addition, meta-analyses are limited in number and confined to only a few noncoding RNAs and cancer types. However, these are required to obtain reliable predicting networks, discriminating between cancer types in which one noncoding RNA may have opposite effects. Unfortunately, information about the sequence of changes in expression profiles of noncoding RNAs at different stages of disease progression is lacking. Finally, we lack algorithms to determine if noncoding RNAs have any clinical value in addition to phenotypic, therapeutic, behavioral, and social data in a predicting model. Artificial intelligence or machine-learning methods may be applied to fit vast amounts of expression data combined with phenotypic, therapeutic, behavioral, and social data.

## 9. Conclusions

Noncoding RNAs regulate the oxidative stress in the tumor microenvironment by regulating mitochondrial function, antioxidant expression, and reprogramming of immune cells. Herein, noncoding RNAs in microvesicles contribute to the communication between different cell types in the tumor microenvironment and inflamed tumor-adjacent tissues. In particular, this communication warrants further investigation.

## Figures and Tables

**Figure 1 cancers-15-01155-f001:**
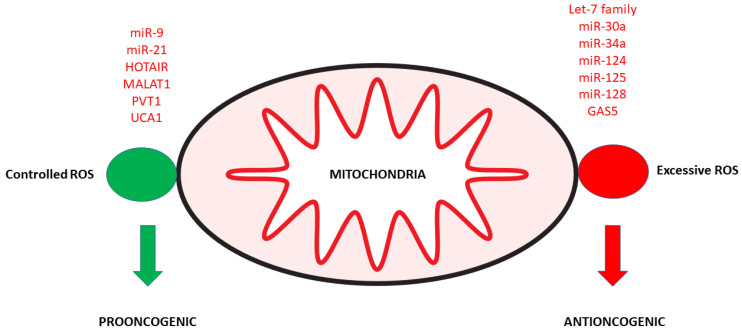
Noncoding RNAs and mitochondrial function and oxidative stress. Controlled release of ROS is required to entertain tumor growth, while excessive ROS production leads to cell death. Upregulated noncoding RNAs are in red.

**Figure 2 cancers-15-01155-f002:**
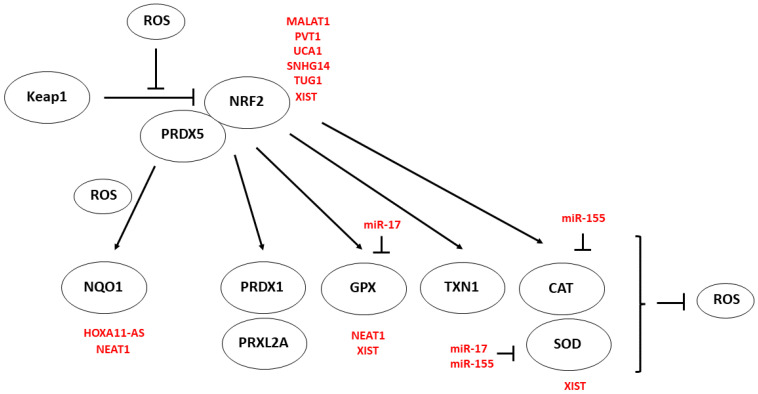
Noncoding RNA and antioxidant activity. ROS levels depend on the tumor’s antioxidant capacity to counterbalance the increased production of ROS. This capacity largely depends on the transcription factor nuclear factor (erythroid-derived 2)–related factor-2 (NRF2). ROS activates tyrosine kinases which dissociate the NRF2 from its inhibitor, the kelch-like ECH-associated protein 1 (Keap1). This dissociation allows the nuclear import of NRF2, activating antioxidant cytoprotective genes, peroxiredoxin-1 (PRDX1), peroxiredoxin-like 2A (PRXL2A), glutathione peroxidases (GPX), thioredoxin-1 (TXN1), and superoxide dismutase (SOD). In addition, NRF2 in ROS-stimulated cancer cells interacts with peroxiredoxin 5 (PRDX5), promoting the expression of NAD(P)H: quinone oxidoreductase 1 (NQO1). NQO1 acts as a superoxide reductase protecting proteins from proteasomal degradation and regulating mRNA translation. Upregulated noncoding RNAs are in red. Arrowheads reflect activation; hammerheads reflect inhibition.

**Figure 3 cancers-15-01155-f003:**
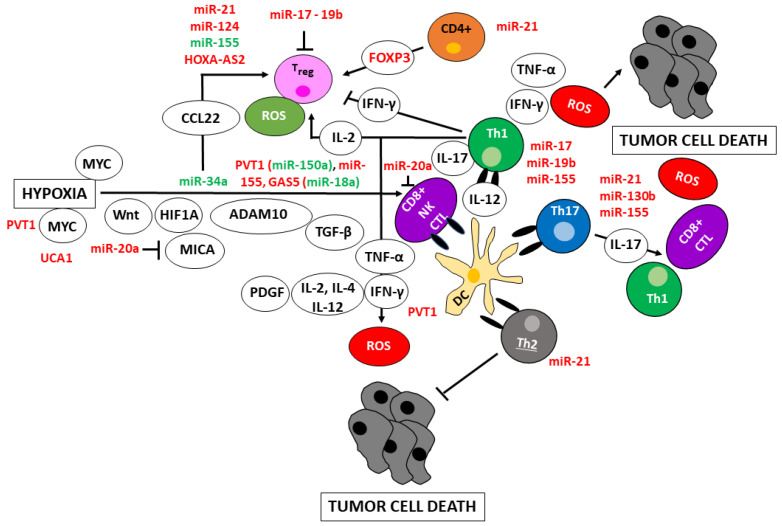
Control of T cell function and oxidative stress by noncoding RNAs. Cancer cells evade immune suppression by downregulating the activity of cytotoxic CD8^+^ T cells and NK cells, suppressing cytokines, and increasing T_reg_ cells. Hypoxia-induced activation of Wnt/β-catenin ablates CD8^+^ T-cell function and suppresses anti-cancer immunity. In addition, hypoxia induces HIF1A, ADAM metallopeptidase domain 10, and MHC class I polypeptide-related sequence A (MICA), which, together with TGF-β, decreases NK cell activity. NK cells share properties with adaptive T cells, and together they induce cancer cell death through IFN-γ and tumor necrosis factor (TNF-α). Thereby, IFN-γ is synergistically enhanced by IL-4, IL-2- and IL-12 and stimulated by platelet-derived growth factor (PDGF)-DD. CD4^+^ T cells consist of T-helper (Th) 1, Th2, Th17, and T_reg_ cells. DC-derived IL-12 favors the differentiation of Th1 cells. Th1 cells produce IL-2, maintaining T_reg_ cells. In contrast, Th2 cells produce IL-4, IL-10, and IL-13, stimulating M2 TAM differentiation. Th17 cells promote tumor growth by IL-17, inducing angiogenesis and recruiting MDSCs. In contrast, Th17 cells cause anti-tumoral immunity by recruiting DCs, CD4^+^ T, and CD8^+^ T cells, activating CD8^+^ T cells and inducing plasticity toward Th1 cells, producing IFN-*γ* and TNF-α, associated with cancer cell death. T_reg_ cells are a subset of FOXP3-positive CD4^+^ T cells strongly inhibiting anti-tumor immune responses mediated by CD4^+^ CD8^+^ T cells. Hypoxia interacts with FOXP3 and enhances the T_reg_ cells through TGF-β1. In aggregate, the activation of cytotoxic T cells, NK cells, and dendritic cells increase ROS. Conversely, the activation of T_reg_ and Th2 cells decreases ROS. The activation of Th1 cells increases ROS by activating dendritic cells but decreases ROS by activating T_reg_ cells. The Th17 cells also have divergent effects by activating MDCs and Th1 cells. Pro-oncogenic ROS are in green, anti-oncogenic ROS are in red. Upregulated noncoding RNAs are in red; down-regulated in green. Arrowheads reflect activation; hammerheads reflect inhibition.

**Figure 4 cancers-15-01155-f004:**
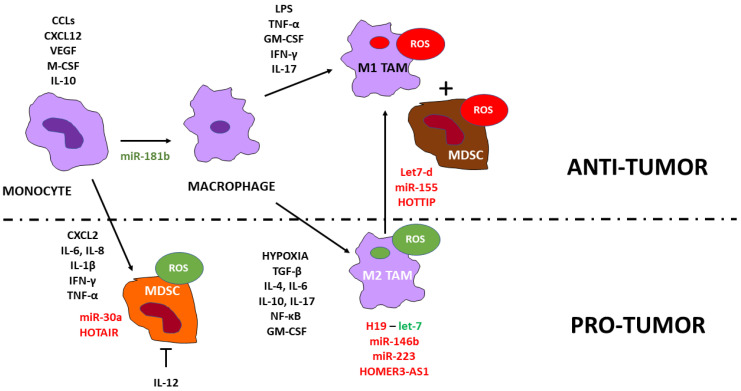
Control of MDSC and macrophage function and oxidative stress by noncoding RNAs. Monocytes and polymorphonuclear cells may differentiate into MDSCs. CXC motif chemokine ligand 2 (CXCL2) released by tumor cells, which in turn activates CXC chemokine receptor 2 (CXCR2), and IL-6, IL-8, and IL-1β recruit tumor-infiltrating MDSCs. In the early stages, NOX2 is the primary source of ROS essential to maintain the undifferentiated state of MDSCs. Cancer cell-derived IL-6 activates MDSCs. MDSCs secrete TGF-β1, inhibiting antigen-specific CD8^+^ T-cell effector functions, increasing T_reg_ cells, inhibiting the DC maturation, and inducing EMT and metastasis. However, in the later stages of tumor growth, the number of MDSCs increases rapidly, along with the production of ROS, NO, and reactive nitrogen species, which may induce death in cytotoxic T cells and T_reg_ cells TAMs. Increased ROS damages adaptive immune response by interfering with IFN-γ expression, hampering activation, viability, and proliferation of T cells. In addition, NO induces T and NK cell apoptosis, increases the p53 tumor suppressor protein, and damages mitochondria. TAMs derive from circulating monocytes recruited by chemokines (CCL2, CCL3, CCL4, CCL5, CCL7, CCL8, CXCL12) and cytokines (VEGF, PDGF, M-CSF, and IL-10). IL-4, IL-6, IL-8, IL-10, IL-11, IL-17, IL-18, IL-33, NF-κB, GM-CSF, TGF-β, and TNF-α induce M2 TAM polarisation. TAMs produce growth factors (HGF, EGF, TGF, and PDGF) and inflammatory cytokines (IL-1β, IL-6, and TNF-α) that can induce EMT in cancer cells. On the other hand, EMT facilitates M2 TAM polarization through GM-CSF, which induces EMT in cancer cells. In addition, TAM-derived TGF-β and TNF-α induce stemness. Finally, M2 TAMs promoted tumor angiogenesis. In contrast, LPS and IFN-γ switch established TAMs into M1 macrophages. Conversely, reprogramming TAMs towards classically activated M1 macrophages induces anti-tumor immunity. Upregulated noncoding RNAs are in red; down-regulated in green. Arrowheads reflect activation; hammerheads reflect inhibition.

**Figure 5 cancers-15-01155-f005:**
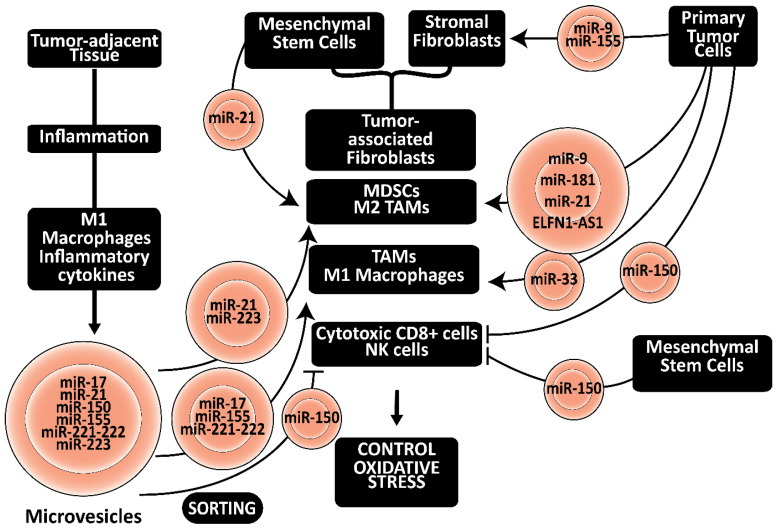
Exchange of noncoding RNAs by microvesicles: Mesenchymal stem cells and primary tumor cells release noncoding RNA-containing microvesicles. These noncoding RNAs control oxidative stress in the tumor environment by regulating the differentiation of fibroblasts to CAFs, and the metabolic reprogramming of T cells, MDSCs, and macrophages. In addition, noncoding RNAs in microvesicles released by inflamed tumor-adjacent tissue may affect the metabolic reprogramming of immune cells in the tumor environment. All indicated noncoding RNAs are enriched in microvesicles. Arrowheads reflect activation; hammerheads reflect inhibition.

**Table 1 cancers-15-01155-t001:** Overview of expected effects of changes in expression of noncoding RNAs.

Noncoding RNA	Observed Changes in Cancer	Possible Mechanism Supporting Tumor Growth	Possible Mechanism Inhibiting Tumor Growth
miR-9	high	Retaining mitochondrial function and protecting against apoptosis; miR-9 in exosomes released by cancer cells induces differentiation of fibroblasts to CAFs	
miR-20a	high	Suppressing NK cell cytotoxicity	
miR-21	high	Protecting mitochondrial function; inducing CD4^+^ T cells and differentiating them to Th2 cells; differentiating Th17 and T_reg_ cells; miR-21 in microvesicles from cancer cells suppresses M1 macrophage polarization; miR-21 in microvesicles secreted by CAFs induces M2 macrophage polarization; miR-21 in microvesicles from macrophages and dendritic cells in tumor-adjacent tissues promotes M2 macrophage polarization in the tumor microenvironment	
miR-130b	high	Activating Th17 cells	
miR-146b	high	Activating M2 TAMs and controlling inflammation	
miR-150	high	Reducing CD8^+^ cytotoxic and NK cells; miR-150 in microvesicles secreted by mesenchymal cells and cancer cells, and tumor-adjacent tissues inhibits NK cell action and improves IL-10 secretion by M2 macrophages	
miR-223	high	Inhibiting and NLRP3 inflammasome, suppressing neutrophil and M1 macrophage polarization; miR-223 in exosomes derived from hypoxic macrophages induces PI3K/AKT signaling, crucial for controlling oxidative stress; miR-223 in microvesicles secreted by tumor-adjacent tissue suppresses the inflammatory tumor microenvironment	
HOTAIR	high	Protecting mitochondrial function; inducing MDSC differentiation; increasing the proportion of M2 macrophages	
MALAT1	high	Protecting mitochondrial function by targeting COX2; increasing antioxidant defense	
UCA1	high	Protecting mitochondrial function; increasing antioxidant defense; attenuated the killing effect of cytotoxic CD8^+^ T cells	
XIST	high	Increasing antioxidant protection	
Let-7b	high		Increasing M1 TAMs
Let-7d	high		Increasing M1 TAMs
miR-16	high		Increasing M1 TAMs
miR-34a	high		Perturbating mitochondrial function; inhibiting recruitment of T_reg_ cells
miR-125	high		Impairing mitochondrial function
miR-128	high		Impairing mitochondrial function and increasing ROS; inducing M1 macrophage polarization
miR-155	high		Reducing antioxidant capacity; increasing CD8^+^ T cells and cytotoxicity of NK cells; promoting the differentiation of Th1 cells and inhibiting differentiation of T_reg_ cells; reprogramming TAMs to pro-inflammatory M1 macrophages; miR-155 in melanoma cell-derived microvesicles induced differentiation of fibroblasts to CAFs; miR-155 in microvesicles secreted by tumor-adjacent tissues exerts anti-tumor effects
GAS5	high		Impairing mitochondrial function; promoting action of NK cells
HOTTIP	high		Inducing polarization of M2 into M1 TAMs
miR-17–19	high	Suppressing the inhibitor of the AKT signaling pathway and activating the AKT pathway; inducing Th17 cell differentiation. However, HOTAIRM1 might downregulate miR-17-5p	Inducing Th1 and suppressing T_reg_ cell differentiation
miR-30a	high	Inducing MDSC differentiation	Inducing mitochondria-dependent apoptosis
miR-124	high	Inducing differentiation of T_reg_ cells	Inducing mitochondria-dependent apoptosis;
NEAT1	high	Increasing antioxidant defense	Promoting the secretion of CD8^+^ T-lymphocyte factors, including TNF-α and IFN-γ
PVT1	high	Protecting mitochondrial function; increasing antioxidant	Enriching CD8^+^ cytotoxic T cell subsets

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
