# Peer review of "Noncoding RNAs Controlling Oxidative Stress in Cancer"

_cancers, 2023, doi:10.3390/cancers15041155_

Round 1

Reviewer 1 Report (Previous Reviewer 1)

Nice work.

Author Response

No further responses

Reviewer 2 Report (Previous Reviewer 2)

The author again presents his review article on the functions of noncoding RNAs on oxidative stress as mediated through the various immune components of cancers. In brief, the author posits that different the expression of noncoding RNAs, as a molecular class, impact the recruitment and function of different components of both tumor cells and recruited immune cells (generally with an impact of inducing a tumor-permissive and inflammatory microenvironment), 

The author has improved many aspects of the manuscript. Most importantly, he has started to introduce context, identifying specific cancers in which the different molecules have specific functions. He has also edited back some of the overlying explanations, allowing for better flow. 

However, there are still issues. 

First and foremost, the author rapidly switches between expression of ncRNAs in "tumors (agnostic of their component cells)," malignant cells, and in the immune cells specifically. There is little explanation of the interplay therein. If an ncRNA is expressed in a malignant cells, how does that lead to impact on the surrounding immune cells (otherwise, you are just describing an association instead of a mechanism). If an ncRNA is expressed in an immune cell, what is driving that expression (again, otherwise you are just describing an association instead of how the cancer induces that expression). Without a link between the two, it is hard to decipher relevance. 

Furthermore, unfortunately there are numerous pathways that are described as being modulated by ncRNAs in a monolithic way (i.e., expression of ncRNA X leads to overexpression of cytokine Y, leading to change in phenotype Z). In reality, there are numerous other pathways involved. For many of the cited examples, the author could simply be identifying associations (mechanism A leads to overexpression of ncRNA B and induction of cytokine C). 

Previously the author stated that he wanted to include all possible molecules to be as unbiased as possible. That is noble, but it leads to a very muddied description. It would, again, be better to focus on a few lead molecules (miR18, PVT1, HOTAIR all seem to be good candidates) and explore how they have been clearly identified mechanistically to impact ROS the way that the author states. 

Also, the title states that the focus is on oxidative stress as related to immune regulation, but well over half of the paper just talks about immune dysregulation, with nearly a whole page on interleukins alone. Many of these molecules have pluripotent effects (MYC, IL's, IFN). If the goal is to discuss oxidative stress, be focused!!

I have spent a significant amount of time reading and re-reading iterations of this manuscript. It is starting to form into something that is manageable and adds to the literature, but right now it it still very cumbersome and not likely to be read broadly. 

Round 2

Reviewer 2 Report (Previous Reviewer 2)

The author has submitted a fourth draft of the manuscript discussing the interplay of noncoding RNAs, reactive oxidative species, and tumor inflammation and microenvironment. The draft has undergone a significant number of changes. The work is more focused on the specific interplay of the proposed components. The addition of the figures and tables now included improve the understanding of the work. There has also been deletion of some of the more questionable parts of the manuscript, such as the roles of the interleukins in the process and if those changes are simply associated with changes in expression of noncoding RNAs or actually caused by them. 

The work is still incredibly dense in its attempt to be all-encompassing. However, the overall concepts are easier to follow. The work may serve as a reference for those investigating ROS and noncoding RNAs. It is reasonably acceptable for publication

This manuscript is a resubmission of an earlier submission. The following is a list of the peer review reports and author responses from that submission.

Round 1

Reviewer 1 Report

In this review article Holvoet et al proposed a set of noncoding RNAs as crucial in regulating energy homeostasis and immune response. Moreover, they are critical in tumors due to their ability to assist in preserving stemness and EMT and defending against apoptosis. In the end, Holvoet et al suggest that ROS levels rely on the tumor's antioxidant capacity to offset the improved production of ROS.

I found the study very interesting and relevant to the field.

I found that this study will provide new insights into the knowledge of non-coding RNAs mediating the regulation of ROS levels.

The article is well written.

I find the conclusion in line with the evidence and arguments presented.

The figures are fine.

References are well-updated.

Minor Suggestion:

Figures 1 to 4 are of good quality with detailed information. However, the meaning of arrowhead (activation) or hammerhead arrows (inhibitions) is missing in all the Figures.

Overall excellent review study!!

Author Response

Dear reviewer,

I thank you for your favorable comments.

I indicated in the legends of the figures that arrowheads mean activation and hammerhead arrows mean inhibitions. 

Reviewer 2 Report

The author has presented a review article on the roles of noncoding RNAs in the control of reactive oxidative species in malignancies. While the concept is interesting, the presentation of the data is too both too expansive (presenting dozens of ncRNAs) and too superficial (statement of how a ncRNA induces or represses a related gene without explanation of mechanism or context). In its current state, the work will not have much value to most readers. 

Generally, I would recommend the author focus more on those ncRNAs with the most supportive data (e.g., MEG3, HOTAIR, H19, MALAT1) and examine the primary data that support their roles. Give more information on how each ncRNA regulates its target, and how these ncRNAs are themselves activated, repressed, or otherwise regulated. Give specific examples from the primary research, including in which cancers there is evidence for each ncRNA. 

This will improve the clarity of the work, the relevance of specific ncRNAs, and the value of the paper overall. 

The abstract itself is very dense, moving from mechanism to mechanisms and proposed molecules rapidly. There is quite a lot of jargon and many abbreviations, not all of which are common (e.g., while most people will interpret ROS as "reactive oxidative species" and not as the proteins ROS1, OXPHOS is not a common abbreviation broadly). Would recommend redrafting the abstract to serve more as a roadmap to the paper overall, with less detail. This is particularly important as, unlike a primary research paper, this work interprets other work and not all the presented references will have the same level of importance or scientific rigor. 

Introduction: a brief review of the mechanism of ROS generation and why ROS generation is useful to cancer cells is merited. Not all readers of this journal may be fully aware of why ROS are crucial. They may be aware of how tumors activate pathways to survive hypoxia, but that would be managing ROS as a toxic compound, not one that is "crucial."

The section on glycolysis vs OXPHOS is too dense and in particular has very little focus on either disease, mechanism, or underlying evidence. See my comments above.

The section on tumor immunity is better delineated. Figure 2 itself is far too dense, and would be better to separate the component immune cells as shown in later figures. However, in many sections it is not clear how the ncRNAs alter the balance of ROS other than indirectly (i.e., most of the discussion is about how the ncRNAs induce a tumor-tolerant immune phenotype, which is not just about ROS balance). 

Finally, the discussion is far too dense for just text. It is not clear at all what message the author is trying to get across. A graphic is necessary to delineate how the proposed "most important ncRNAs (my words, not his)" actually work and in which contexts. 

With some significant editing, this could be an interesting manuscript, but generally not useful in its current form

Author Response

Response to the comments of reviewers:

Referee 1:

I thank you for your favorable comments.

I indicated in the legends of the figures that arrowheads mean activation and hammerhead arrows mean inhibitions. 

Referee 2:

Comment 1: Generally, I would recommend the author focus more on those ncRNAs with the most supportive data (e.g., MEG3, HOTAIR, H19, MALAT1) and examine the primary data that support their roles. Give more information on how each ncRNA regulates its target, and how these ncRNAs are themselves activated, repressed, or otherwise regulated. Give specific examples from the primary research, including in which cancers there is evidence for each ncRNA. This will improve the clarity of the work, the relevance of specific ncRNAs, and the value of the paper overall. 

Response: we preferred to perform an unbiased literature search to identify noncoding RNAs with an experimental basis for a regulatory role in tumor oxidative stress by affecting energy homeostasis and immune response. There are pieces of evidence that miR-155, GAS5, H19, MALAT1, NEAT1, PVT1, TUG1, UCA1, XIST1, and lncRNAs from the small nucleolar RNA host gene family and the homeobox A and B cluster regulate energy homeostasis and immune response to retain mitochondrial function, immune cells which facilitate glycolysis, and cancer cell survival. Their relationship with stemness, epithelial-mesenchymal transition, and apoptosis further underlie their importance in cancer. Furthermore, experimental evidence supported common paths in regulating this cluster of noncoding RNAs, mainly by hypoxia, MYC, TGF-β, and oxidative stress and inflammation. In addition, they may control ROS levels by regulating the tumor's antioxidant capacity to counterbalance the increased production of ROS. Finally, exosomal enrichment allows communication between cell types in a tumor and between a tumor and its environment.

An a priori selection of noncoding RNAs may give a more straightforward image that is not wholly accurate. In addition, we carefully selected references, including the complete primary data.

We added a paragraph stressing that the quality of a review paper depends on the quality of the primary papers. Therefore, we mainly included a role of a noncoding RNA when several papers confirmed this role. In addition, we noted several shortcomings.

“However, note that the quality of a review paper depends on the quality of the primary papers. Therefore, we mainly included a role of a noncoding RNA when several papers confirmed this role. Notwithstanding, we noticed significant shortcomings in noncoding RNA-related research. First, functional and clinical studies are mostly limited to a single noncoding RNA, although noncoding RNAs function in networks. The outcome of silencing a single lncRNA is attributed chiefly to one miR's effect, although no broader omics search is performed to unravel all potentially involved miRs. In addition, the nonspecific effects of silencing experiments are mostly disregarded, and reconstitution experiments are not performed. This may explain contrasting results. We hope this review paper warrants further investigation of the cluster as a whole, given the overlap in regulation of the noncoding RNAs.’

We added a new part on regulation and included a new figure 5.

Comment 2: The abstract itself is very dense, moving from mechanism to mechanisms and proposed molecules rapidly. There is quite a lot of jargon and many abbreviations, not all of which are common (e.g., while most people will interpret ROS as "reactive oxidative species" and not as the proteins ROS1, OXPHOS is not a common abbreviation broadly). Would recommend redrafting the abstract to serve more as a roadmap to the paper overall, with less detail. This is particularly important as, unlike a primary research paper, this work interprets other work and not all the presented references will have the same level of importance or scientific rigor.

Response: We redrafted the abstract explaining steps in our literature search.

‘A controlled release of ROS in response to hypoxia induces tumor growth. However, high ROS levels initiate oxidative stress and cancer cell death. Metabolic reprogramming to protect mitochondria is one way to control ROS to support tumor growth. ROS levels also depend on the activity of cytotoxic T and natural killer cells, anti-immune regulatory T cells, myeloid-derived suppressor cells, and tumor-associated M2 macrophages. We performed an unbiased literature search to identify noncoding RNAs with an experimental basis for a regulatory role in tumor oxidative stress by affecting energy homeostasis and immune response. There are pieces of evidence that miR-155, GAS5, H19, MALAT1, NEAT1, PVT1, TUG1, UCA1, XIST1, and lncRNAs from the small nucleolar RNA host gene family and the homeobox A and B cluster regulate energy homeostasis and immune response to retain mitochondrial function, immune cells which facilitate glycolysis, and cancer cell survival. Their relationship with stemness, epithelial-mesenchymal transition, and apoptosis further underlie their importance in cancer. Furthermore, experimental evidence supported common paths in regulating this cluster of noncoding RNAs, mainly by hypoxia, MYC, TGF-β, and oxidative stress and inflammation. In addition, they may control ROS levels by regulating the tumor's antioxidant capacity to counterbalance the increased production of ROS.  Finally, exosomal enrichment allows their communication between cell types in a tumor and between a tumor and its environment.

 Comment 3: Introduction: a brief review of the mechanism of ROS generation and why ROS generation is useful to cancer cells is merited. Not all readers of this journal may be fully aware of why ROS are crucial. They may be aware of how tumors activate pathways to survive hypoxia, but that would be managing ROS as a toxic compound, not one that is "crucial."

We have expanded the part about glycolysis and ROS.

‘Controlled reactive oxygen species (ROS) levels are crucial for tumor growth in hypoxia 1. However, too high ROS levels cause oxidative stress-induced cancer cell death. In order to control ROS, cancer cells undergo metabolic reprogramming, preferring glycolysis (Warburg effect) above oxidative phosphorylation (OXPHOS) 2. Normal cells under aerobic conditions utilize glucose to produce pyruvate, which then gets oxidized in the mitochondria via the tricarboxylic acid (TCA) cycle into carbon dioxide, a process called oxidative phosphorylation (37). Oxidation of pyruvate generates 36 molecules of ATP in mitochondria. Nicotinamide adenine dinucleotide (NADH) is generated from the TCA cycle, and ATP is produced most efficiently through the phosphorylation of ADP, accompanied by the transformation of NADH to NAD+ through the electron transfer chain in the mitochondrial inner membrane. This process consumes oxygen and produces CO2 from the breakdown of pyruvate. When the oxygen supply is insufficient, pyruvate tends to be metabolized to lactate in anaerobic glycolysis; this process generates lactate and two molecules of ATP, independent of the TCA cycle during mitochondrial respiration 3.  However, cancer cells produce energy via the conversion of glucose into lactate, despite the presence of oxygen, a process known as aerobic glycolysis. This Warburg effect has been demonstrated to benefit proliferative cancer cells through rapid ATP generation and simultaneous flux through the pentose phosphate pathway to support redox homeostasis and biosynthesis. Indeed it decreases mitochondrial ROS, protects against apoptosis, increases cancer cell proliferation, and inhibits tumor growth without apparent toxicity 4. While the Warburg effect is commonly seen in proliferative cancer cells, metabolic phenotypes between proliferative primary cancer cells and disseminated cancer cells are not the same 5 [3]. Anaerobic glycolysis, in contrast, is initiated under hypoxic conditions in normal cells or when cancer cells adapt to hypoxia 6. However, even then, long-term hypoxia causes mitochondrial stress, generating intolerable levels of ROS and inducing apoptosis 7.’

Comment 4: The section on glycolysis vs OXPHOS is too dense and in particular has very little focus on either disease, mechanism, or underlying evidence. See my comments above.

Response: see above (comment 3).

Comment 5: The section on tumor immunity is better delineated. Figure 2 itself is far too dense, and would be better to separate the component immune cells as shown in later figures. However, in many sections it is not clear how the ncRNAs alter the balance of ROS other than indirectly (i.e., most of the discussion is about how the ncRNAs induce a tumor-tolerant immune phenotype, which is not just about ROS balance). 

Response: we simplified the figure and indicated which cells appear to produce oncogenic levels of ROS (green), and which antioncogenic levels of ROS (red).

Comment 6: Finally, the discussion is far too dense for just text. It is not clear at all what message the author is trying to get across. A graphic is necessary to delineate how the proposed "most important ncRNAs (my words, not his)" actually work and in which contexts. 

Response: we have rewritten the discussion and added a new figure 6 showing the overlap in additional effects of the noncoding RNAs.

Comment 7: With some significant editing, this could be an interesting manuscript, but generally not useful in its current form.

Response: We hope the modifications and additions make this an interesting and useful paper.

Round 2

Reviewer 2 Report

I appreciate the author's attempt to address the issues I originally raised. The author stated very clearly in the response that they attempted to present an unbiased review of noncoding RNAs, both long ncRNAs and microRNAs. 

That said, the resulting manuscript remains problematic. While the author states that, to best avoid bias, it is best to report on multiple ncRNAs, that is inherently not possible. The sum total of ncRNAs remains likely still unknown, and direct and indirect effects also remain unexplored. The author states that they included ncRNAs when there were multiple papers supporting a role, but that is not actually true. For each ncRNA cited in a given paragraph, there is generally 1 citation identified. What is more accurate is that, for each pathway (e.g., PDK1, MYC), there are multiple ncRNAs identified and cited. 

Additionally, there is absolutely no context provided for any of the ncRNAs. Not all ncRNAs are expressed in any single cancer or across all cancers, just as not all oncogenes are expressed in all cancers. For example, there are many cancers in which MYC (or its paralogs) are not expressed. Similarly, the author states that "Cancer cells secrete IL-6" when that is again very context specific. A casual reader or one not well-educated on these topics (e.g, students) may be mislead by such statements, and any such review paper can do more harm than good. 

There are also structural problems to the paper. There is an introduction that is dense, followed by a review of oxidative phosphorylation vs Warburg and in the context of ROS. Section 3 then just jumps into specific lncRNAs and how they interact with pathways that regulate ROS, OXPHOS and the Warburg effect, with no rationale, moving from one activated gene to another. In the immune section, there is significant back and forth in each section; in the T-cell section, significant effort is spent discussing MDSCs and macrophages, and even the discussion of ncRNAs and T-cells includes indirect effects on T-cells. There is then additional discussion of just the MDSCs and TAMs alone. 

No review paper can truly comprehensively review all topics. As you state, the review is only as good as the quality of the primary papers. Stating so is not exculpatory. The reviewer's job is to evaluate those papers, identify those with best data and justification, and to present those. 

In my honest opinion, I can only support publication of this manuscript if the author were to put forth a true good-faith effort to provide better context for the evidence of when specific pathways are activated. Broad statements like "Cancer cells do X" must either be given with either justification (e.g., the Warburg effect is activated in many cancers with good evidence from multiple labs) or with narrowed context (e.g., in which cancers has HOTAIR been shown to have effects; in which cancers is IL6 or IL10 expressed). In its current form, it does not signficantly add to the great discussion of cancer research, ncRNA research, or the roles of ROS and related immune functions in cancer. I unfortunately cannot support publication.